# In Vitro Activity of “Old” and “New” Antimicrobials against the *Klebsiella pneumoniae* Complex

**DOI:** 10.3390/antibiotics13020126

**Published:** 2024-01-26

**Authors:** Alicja Sękowska

**Affiliations:** 1Microbiology Department, Ludwik Rydygier Collegium Medicum in Bydgoszcz, Nicolaus Copernicus University in Torun, 9 Maria Skłodowska-Curie St., 85-094 Bydgoszcz, Poland; asekowska@cm.umk.pl; Tel.: +48-52-585-44-80; 2Clinical Microbiology Department, Dr. A. Jurasz University Hospital No. 1, 85-094 Bydgoszcz, Poland

**Keywords:** ceftazidime–avibactam, meropenem–vaborbactam, colistin, fosfomycin, *K. pneumoniae*, *K. variicola*

## Abstract

The *Klebsiella pneumoniae* complex is a commonly isolated bacteria in human infections. These opportunistic pathogens pose a serious threat to public health due to their potential transmission to the human population. Resistance to carbapenems is a significant antimicrobial resistance mechanism, leading to limited therapeutic options. Therefore, the aim of this study was to evaluate the in vitro activity of fosfomycin, colistin, ceftazidime–avibactam, and meropenem–vaborbactam against multidrug-resistant *K. pneumoniae* complex strains. This study involved 160 strains of Gram-negative rods, comprising 138 *K. pneumoniae* and 22 *K. variicola*. The minimal inhibitory concentration of fosfomycin was estimated using the agar dilution method, and for colistin, the microdilution method was employed. Susceptibility to ceftazidime–avibactam and meropenem–vaborbactam was determined using the gradient strip method. All analyzed *K. pneumoniae* complex isolates produced extended-spectrum β-lactamases, and 60.0% exhibited carbapenemases. The majority of the analyzed strains were susceptible to fosfomycin and colistin (62.5%). Among pandrug-resistant *K. pneumoniae* complex isolates, the highest susceptibility was observed with colistin (43.9%). Fosfomycin demonstrated good activity against ESβLs- and VIM-positive isolates from this complex. Colistin also exhibited satisfactory in vitro activity against VIM- and KPC-positive isolates from the *K. pneumoniae* complex. Ceftazidime–avibactam displayed good activity against *K. pneumoniae* complex strains producing ESβLs, KPC, and OXA enzymes. Additionally, meropenem–vaborbactam showed satisfactory in vitro activity against ESβLs- and KPC-positive isolates from this complex.

## 1. Introduction

The *Klebsiella pneumoniae* complex is a common causative bacteria for both community- and hospital-acquired infections. The predominant and frequently isolated species within this complex is *K. pneumoniae*. The incidence of these bacteria causing severe infections with substantial morbidity and mortality has been steadily rising [1]. Notably, in recent years, *Klebsiella variicola* has been increasingly identified, particularly in cases of bloodstream infections [2,3]. Both *K. pneumoniae* and *K. variicola* are associated with severe infections leading to high mortality rates, especially in immunocompromised patients [1,4]. Furthermore, these bacteria frequently exhibit diverse mechanisms of antimicrobial resistance. For many years, the predominant and critical mechanism of antimicrobial resistance within the *K. pneumoniae* complex was the production of extended-spectrum β-lactamases (ESβLs). However, in the past decade, more and more reports have appeared describing the identification of strains producing carbapenemases [5]. Strains producing EsβLs and carbapenemases can colonize the gastrointestinal tract of healthy individuals, subsequently leading to serious infections. Within the *K. pneumoniae* isolates, three classes of carbapenemases are identified: class A (*Klebsiella pneumoniae* carbapenemases, KPC), class B (metallo-β-lactamases, MβL), and class D (oxacillinases, OXA) [6]. Strains producing KPC enzymes are frequently isolated in the United States and Israel [7]. In Poland, the first KPC-positive *K. pneumoniae* strain was isolated in 2008. Between 2019 and 2021, the frequency of isolation increased by around 200% and accounted for 18% of all cases. [8,9]. The first MβL-positive *K. pneumoniae* strain in Poland was isolated in 2006 and identified as the VIM-1 enzyme [10]. Since the isolation of the first strain in Poland, VIM enzymes have been identified in various species, with the two dominant ones being *Enterobacter hormaechei* and *Klebsiella oxytoca*. However, in recent years, there has been a shift, and an increase in the frequency of isolation of *K. pneumoniae* and *Escherichia coli* strains has been observed [9]. Notably, VIM-positive strains were the only carbapenemase-producing isolates for which no increase was recorded in Poland in 2020–2021. The first NDM-positive strain appeared in Poland in 2012 and rapidly spread to other hospitals, causing local epidemics. In 2021, NDM-positive strains accounted for 73% of all carbapenemase-producing Enterobacterales strains in Poland, reflecting a substantial increase in incidence (of 99%) compared to 2019 [9]. In Poland, the first OXA-48-positive isolate, identified as *E. hormaechei* subsp. *steigerwaltii*, was recognized in 2012. Until 2017, OXA-48-positive strains were relatively infrequently isolated in Poland. However, in 2021, there was an observed increase in the frequency of strains with this phenotype by approximately 70% compared to 2019. The OXA-48-like group of carbapenemases includes various enzymes such as OXA-48, OXA-181, OXA-232, OXA-204, OXA-162, and OXA-244 [11]. Although carbapenemases can be produced by all Enterobacterales, *K. pneumoniae* remains the primary species in which they are identified. According to data from the European Antimicrobial Resistance Surveillance System (EARSS), in Poland, the proportion of *K. pneumoniae* strains resistant to carbapenems isolated from invasive infections increased from 0.8% in 2012 to 6.4% in 2017 and further rose to 16.8% in 2022 [5]. The prevalence of carbapenem-resistant *K. pneumoniae* strains in Europe varies, ranging from 0.0% in Finland and Iceland to 72.0% in Greece, according to EARSS data from 2022 [5]. Additionally, strains producing more than one β-lactamase from different groups are increasingly being identified. These include combinations such as ESβLs and carbapenemases or two different carbapenemases. In such scenarios, therapeutic options become highly limited. Information regarding the antimicrobial resistance profile of *K. variicola* strains is scant, often limited to case reports.

Fosfomycin, colistin, ceftazidime–avibactam, and meropenem–vaborbactam are registered for treating severe infections caused by strains with limited therapeutic alternatives. The inclusion of these antimicrobials is justified for various reasons. Fosfomycin and colistin possess unique and distinct mechanisms of action, exhibiting bactericidal activity against a broad spectrum of bacterial species. Additionally, fosfomycin demonstrates excellent diffusion in body tissues and has low toxicity. While monotherapy with fosfomycin is not recommended for treating infections caused by MβL-producing *Enterobacterales*, it exhibits synergy with other antimicrobials, such as ceftazidime–avibactam [12,13]. On the other hand, despite its high toxicity, colistin remains a drug of last resort for treating multidrug-resistant (MDR) bacterial infections, including those caused by *K. pneumoniae* [14,15]. The introduction of new combinations of β-lactams with β-lactamase inhibitors has significantly broadened therapeutic options for treating infections caused by carbapenemase-producing strains. These drugs are deemed safe for patients and exhibit effective tissue penetration. Avibactam, a diazabicyclooctane non-β-lactam, binds covalently and reversibly to β-lactamases. Its crucial advantage lies in its ability to inhibit ESβLs, AmpC β-lactamases, as well as class A and D carbapenemases [16]. Vaborbactam, the first boronic acid β-lactamase inhibitor, belongs to a group known for the reversible and competitive inhibition of serine β-lactamases. It effectively inhibits class A β-lactamases, including KPC carbapenemases, and class C β-lactamases. However, it does not inhibit class B and class D β-lactamases [16].

The objective of this study was to compare the in vitro activity of fosfomycin, colistin, ceftazidime–avibactam, and meropenem–vaborbactam against MDR *K. pneumoniae* complex, encompassing isolates producing ESβLs and various carbapenemases.

## 2. Results

### 2.1. Isolates Collection

The study encompassed 160 *K. pneumoniae* complex strains, comprising 138 (86.2%) *K. pneumoniae* and 22 (13.8%) *K. variicola*. Among these, 146 (91.2%) were obtained from clinical specimens (125 *K. pneumoniae* and 21 *K. variicola*), while 14 (8.8%) were isolated from gastrointestinal colonization samples (stool and rectal swab) (13 *K. pneumoniae* and 1 *K. variicola*). A total of 72 (45.0%) of the analyzed strains were isolated as monocultures (63 *K. pneumoniae* and 9 *K. variicola*).

### 2.2. Identification of Strains and Antimicrobial Susceptibility Testing

In the mass spectrometry method, all of the *K. pneumoniae* complex strains exhibited identification values over 2.300, which means reliable identification at the genus and species level.

All of the analyzed *K. pneumoniae* complex strains were MDR. Among the analyzed isolates, 41 (25.6%) were PDR, and they all belonged to *K. pneumoniae* species. In the group of PDR strains, 39 (95.1%) produced different carbapenemases and only 2 (4.9%) produced ESβLs enzymes. The detailed data of on susceptibility of *K. pneumoniae* complex strains to antimicrobials, obtained from Phoenix, are presented in Appendix A.

### 2.3. Detection of Extended-Spectrum β-Lactamases and Carbapenemases

In the automated method, ESβL production was confirmed for all *K. pneumoniae* strains (*n* = 138), and in a double-disc test, it was confirmed for all *K. variicola* strains (*n* = 22). In the eazyplex^®^ SuperBug CRE test, 140 of the analyzed isolates produced ESβLs, with 139 strains belonging to the CTX-M1 group and 1 strain producing two ESβLs from groups CTX-M1 and CTX-M9. For 115 of the analyzed isolates (113 *K. pneumoniae* and 2 *K. variicola*) resistant to at least one carbapenem, the CIM test was performed. Among these, 96 (60.0%) of *K. pneumoniae* strains tested positive in the CIM test, indicating the production of carbapenemases. The distribution of carbapenemases included NDM in 40 strains, KPC in 17, VIM in 15, OXA-181 in 7, OXA-48 in 2, and 15 strains producing more than one carbapenemase (multi-carbapenemase strains). None of the *K. variicola* isolates produced carbapenemases. Detailed data on *K. pneumoniae* complex strains producing β-lactamases are presented in Table 1.

### 2.4. MIC Determination

Among the 160 *K. pneumoniae* complex strains, 100 (62.5%) were susceptible to fosfomycin and colistin, 99 (58.1%) to ceftazidime–avibactam, and 88 (55.0%) to meropenem–vaborbactam. The detailed susceptibility profiles for the specific groups of the analyzed *K. pneumoniae* complex strains are presented in Figure 1.

In the group of PDR *K. pneumoniae* isolates (41 isolates), the majority were susceptible to colistin (18, 43.9%), followed by fosfomycin (10, 24.4%), ceftazidime–avibactam (10, 24.4%), and meropenem–vaborbactam (6, 14.6%).

Among the carbapenemase-positive strains (96 isolates), the majority were susceptible to colistin (60, 62.5%), followed by fosfomycin (47, 49.0%), ceftazidime–avibactam (35, 36.5%), and meropenem–vaborbactam (29, 30.2%).

The *K. variicola* ESβLs-positive strains exhibited the lowest MIC_50_ and MIC_90_ values for all tested antimicrobials, with the exception of the MIC_50_ for colistin, which had the highest value among all analyzed groups. Detailed data on MIC_50_ and MIC_90_ values are provided in Table 2.

Fosfomycin, colistin, ceftazidime–avibactam, and meropenem–vaborbactam MICs distributions for the specific groups of strains are presented in Table 3, Table 4 and Table 5.

The MICs of fosfomycin ranged from 4 to >256 mg/L for PDR, carbapenemase-positive, and ESβLs-positive *K. pneumoniae* isolates, and from 4 to 128 mg/L for *K. variicola* ESβLs-positive isolates.

The MICs of colistin ranged from 0.25 to >16 mg/L for PDR and carbapenemase-positive *K. pneumoniae* isolates, from 0.25 to 16 mg/L for ESβLs-positive *K. pneumoniae* isolates, and from 0.25 to 8 mg/L for *K. variicola* ESβLs-positive isolates.

The MICs of ceftazidime–avibactam ranged from 0.25 to 256 mg/L for PDR isolates, from 0.38 to 256 mg/L for carbapenemase-positive *K. pneumoniae* isolates, from 0.047 to 64 mg/L for ESβLs-positive *K. pneumoniae* isolates, and from 0.047 to 2 mg/L for *K. variicola* ESβLs-positive isolates. The MICs of meropenem–vaborbactam ranged from 0.064 to 256 mg/L for PDR strains, from 0.032 to >256 mg/L for carbapenemase-positive strains, from 0.032 to >256 mg/L for ESβLs-positive *K. pneumoniae* strains, and from 0.032 to 2 mg/L for *K. variicola* ESβLs-positive strains.

## 3. Materials and Methods

### 3.1. Bacterial Strains

The study included 160 non-replicated strains of the *K. pneumoniae* complex isolated from various clinical samples derived from patients at Dr. A. Jurasz University Hospital No. 1 in Bydgoszcz, Poland. These isolates were collected over a one-year period, spanning from 1 October 2022 to 30 September 2023. The strains were obtained through standard diagnostic procedures at the Microbiology Department of Dr. A. Jurasz University Hospital No. 1 in Bydgoszcz, Poland.

### 3.2. Identification and Susceptibility Testing

Isolate identification was performed using matrix-assisted laser desorption ionization time-of-flight mass spectrometry (MALDI-TOF MS) on the MALDI TOF Biotyper Microflex LT/SH system (Bruker, Bremen, Germany) with version 7.0.0.1 software. For strains with a value ≥2.300, identification was conducted once, while for isolates with values between 2.000 and 2.299, identification was performed thrice.

Antimicrobial susceptibility tests for all the analyzed strains were conducted using the Phoenix M50 system (Becton Dickinson, NJ, USA) with NMIC-408 panels. The obtained results were automatically interpreted following the European Committee on Antimicrobial Susceptibility Testing (EUCAST) 2023 Recommendations (v 13.0) [17]. An isolate was classified as MDR if it demonstrated non-susceptibility to ≥1 agent in >3 antimicrobial categories and as pandrug-resistant (PDR) if it exhibited non-susceptibility to all antimicrobial agents [18]. The definitions of MDR and PDR were determined based on the results obtained from Phoenix.

### 3.3. Detection of Enzymes

For the detection of ESβLs enzymes in *K. pneumoniae* strains, the Phoenix M50 System (Becton Dickinson, NJ, USA) with NMIC-408 panels was employed. Meanwhile, for *K. variicola* isolates, ESβLs enzyme activities were identified using the disc diffusion method, specifically the double-disc synergy test with ceftazidime (30 μg), cefotaxime (30 μg), cefepim (30 μg), and amoxicillin with clavulanic acid (30 μg) (Liofilchem, Abruzzi, Italy). Control strains, including *E. coli* ATCC 25922 (ESβLs-negative) and *K. pneumoniae* ATCC 700603 (ESβLs-positive), were utilized. For strains resistant to carbapenems, both the Carbapenem Inactivation Method [19] and eazyplex^®^ SuperBug CRE test (Amplex Diagnostics, Gars-Bahnhof, Germany) were employed. Reference strains for this analysis included *K. pneumoniae* NCTC 13442 (OXA-positive), *K. pneumoniae* NCTC 13438 (KPC-positive), *K. pneumoniae* BAA-2146 (NDM-positive), *K. pneumoniae* NCTC 13440 (VIM-positive), and *K. pneumoniae* ATCC 700603 (ESβLs-positive).

### 3.4. MIC Determination

The determination of the fosfomycin MIC was carried out by dilution in agar using AD Fosfomycin (Liofilchem, Abruzzi, Italy). Fosfomycin concentrations in agar ranged from 0.25 mg/L to 256 mg/L. Breakpoints for fosfomycin were interpreted according to EUCAST Recommendations 2023 (v. 13.0) [17] (MIC ≤ 32 mg/L—susceptible, >32 mg/L—resistant). For colistin, MIC was determined using the microdilution method with the MIC COL test (Inc. Diagnostics, Galanta, Slovakia). Colistin concentrations in the broth ranged from 0.25 mg/L to 16 mg/L. Breakpoints for colistin were interpreted according to EUCAST Recommendations 2023 (v. 13.0) [17] (MIC ≤ 2 mg/L—susceptible; >2 mg/L—resistant). Susceptibility breakpoints for ceftazidime–avibactam and meropenem–vaborbactam were established by the gradient strip method (Liofilchem, Abruzzi, Italy). Ceftazidime–avibactam and meropenem–vaborbactam concentrations in the strip ranged from 0.016 mg/L to 256 mg/L. Breakpoints for ceftazidime–avibactam and meropenem–vaborbactam were interpreted according to EUCAST Recommendations 2023 (v. 13.0) [17] (MIC ≤ 8 mg/L—susceptible; >8 mg/L—resistant). *E. coli* ATCC 25922 and *E. coli* NCTC 13846 (mcr-1 positive) were used as reference strains.

## 4. Discussion

The escalating issue of bacterial resistance to antimicrobials has prompted the exploration of new drugs and the reconsideration of well-established ones. Fosfomycin and colistin, discovered a few decades ago, have regained attention. The introduction of intravenous fosfomycin in 2019, particularly in European countries like Poland, broadened its utility against infections caused by MDR strains. Colistin, despite its historical use, faced limitations due to toxicity after 1970. However, both drugs offer a broad spectrum of activity, encompassing strains producing β-lactamases from classes A to D. While various studies have addressed the activity of fosfomycin and colistin, a notable gap exists concerning isolates from Eastern Europe, including Poland. The results of this study demonstrated that over 62% of the analyzed *K. pneumoniae* complex strains in Poland were susceptible to fosfomycin. Notably, a prior study by Kowalska-Krochmal et al. [20] on a substantial cohort of *Klebsiella* spp. isolates (250 strains) from patients with invasive infections in Poland reported a 66.0% susceptibility to fosfomycin, with MIC_50_ at 32 mg/L and MIC_90_ at 512 mg/L. In the analyzed group, 86 isolates were ESβLs-positive, 26 were ESβLs- and carbapenemase-positive, and 58 were carbapenemase-positive. The results of this study indicated that fosfomycin exhibited high potency, with over 86% susceptibility, against *K. pneumoniae* VIM-positive isolates and *K. variicola* ESβLs-positive isolates. These findings underscore the importance of considering regional variations in antimicrobial susceptibility, and obtained results contribute valuable insights into the efficiency of fosfomycin against specific strains in Poland.

According to research conducted in Turkey [21], 53.2% of *Klebsiella* spp. isolates were susceptible to fosfomycin. Notably, the highest susceptibility was observed for OXA-48- and NDM-positive strains, with 73.3% being susceptible, while strains producing only NDM enzymes exhibited the lowest susceptibility at 33.3%. In this study, none of the nine analyzed OXA-positive isolates were susceptible to fosfomycin, with only two isolates producing OXA-48 and seven strains producing OXA-181. In contrast, a study by Demirci-Duarte et al. [21] reported that 50.5% of OXA-48-positive *Klebsiella* spp. isolates (*n* = 104) were susceptible to fosfomycin. Additionally, Aprile et al. [22] found that 76% of KPC-positive *K. pneumoniae* strains were susceptible to fosfomycin, while none of the NDM- and OXA-48-positive strains showed susceptibility. However, in this study, the value for KPC-positive strains was slightly lower, around 64%, although the number of strains analyzed was smaller. A further study from Latin America [23] reported varying susceptibility of *K. pneumoniae* strains to fosfomycin, ranging from 90.9% for strains isolated in Chile to 100% for strains isolated in Mexico, with a total of 601 *K. pneumoniae* isolates included. For carbapenem-non-susceptible *K. pneumoniae* strains (183), the susceptibility values were 71.4% and 100%, respectively. On the other hand, Zarakolu et al. [24] reported that 90.7% of carbapenem-susceptible *K. pneumoniae* isolates and 69.4% of carbapenem-resistant isolates were susceptible to fosfomycin.

The results of this study revealed that over 62% of the analyzed *K. pneumoniae* complex strains were susceptible to colistin. Comparable findings were reported in Greece [25], where the antimicrobial activity of colistin demonstrated a susceptibility rate of 64% among 392 carbapenem-resistant *K. pneumoniae* strains. In Poland, Pruss et al. [26] conducted a study on a group of 200 *K. pneumoniae* isolated from clinical samples. Among the analyzed ESβLs-positive strains, they were all susceptible to colistin. For carbapenemase-positive isolates, the susceptibility varied based on the phenotype. The highest susceptibility to colistin was obtained for KPC-positive isolates (100%), followed by NDM-positive isolates (82.3%), and only 50.0% for OXA-48-positive strains. The results of this study showed that colistin was highly potent (≥80% susceptibility) against KPC-positive and VIM-positive *K. pneumoniae* isolates. However, the lowest susceptibility to colistin was noted for OXA-positive strains, with only 33.4% susceptibility, including the OXA-48 and OXA-181 strains. Research conducted in European countries, including Poland [27], reported that 95.6% of *K. pneumoniae* among 4201 isolates were susceptible to colistin, with MIC_50/90_ values of 1 mg/L, ranging from ≤0.12 mg/L to >4 mg/L. Another study from Spain [28] found that the susceptibility of carbapenemase-producing *K. pneumoniae* strains to colistin decreased from 86.5% to 68.3% over three years. Interestingly, a substantial difference in colistin MIC_90_ values was observed among carbapenemase-producing *K. pneumoniae* isolates in Greece, India, and Poland, with values of 64 mg/L, 32 mg/L, and 8 mg/L, respectively. Meanwhile, colistin MIC_50_ values were similar, at 1 mg/L, 0.5 mg/L, and 0.75 mg/L, respectively.

There are no data in the literature regarding the frequency of the occurrence of colistin-susceptible *K. variicola* isolates. However, a few articles describe *K. variicola* strains resistant to this antimicrobial [29,30]. In a previous study, among 13 *K. variicola* strains isolated from clinical samples, 46.1% exhibited resistance to colistin (unpublished data).

Ceftazidime–avibactam and meropenem–vaborbactam are combinations of established β-lactams with new β-lactamase inhibitors. These antimicrobials received approval from the US Food and Drug Administration and the European Medicines Agency several years ago. Both ceftazidime–avibactam and meropenem–vaborbactam exhibit activity against strains producing class A and C β-lactamases. Additionally, ceftazidime–avibactam demonstrates efficacy against strains producing class D β-lactamases.

The results of this study demonstrate that ceftazidime–avibactam and meropenem–vaborbactam were highly effective (>88% susceptibility) against *K. pneumoniae* KPC-positive isolates, as well as *K. variicola* and *K. pneumoniae* ESβLs-positive isolates. Previous studies from this medical center [31] have also confirmed the high in vitro activity of ceftazidime–avibactam against *K. pneumoniae* strains producing ESβLs enzymes. In the cited study, out of four strains producing carbapenemases (two OXA-48 and two KPC), three were susceptible to ceftazidime–avibactam. A study from Latin America [23] found that the susceptibility of *K. pneumoniae* strains to ceftazidime–avibactam varied from 87.0% for strains isolated in Brazil to 100% for strains isolated in Mexico. For carbapenem-non-susceptible *K. pneumoniae* strains (*n* = 183), the values were 83.3% and 100%, respectively, with the lowest value obtained in Colombia at 68.6%. According to research conducted in European countries, including Poland [27], 98.9% of *K. pneumoniae* of 4201 isolates were susceptible to ceftazidime–avibactam with MIC_50_ 0.12 mg/L and MIC_90_ 1 mg/L, ranging from ≤0.015 mg/L to >128 mg/L. Another study from Pittsburgh [32] noted that the susceptibility of *K. pneumoniae* strains to ceftazidime–avibactam achieved 79.0%. All of the analyzed isolates were carbapenem-resistant, and 93% of them produced KPC enzymes. In this study, over 35% of carbapenemase-positive *K. pneumoniae* strains were susceptible to ceftazidime–avibactam. Among KPC-positive strains, all were susceptible to this drug. Additionally, a high percentage of OXA-positive *K. pneumoniae* strains were susceptible to ceftazidime–avibactam, almost 89%. In contrast, Bianco et al. [33] analyzed susceptibility to selected antimicrobials in seven multi-carbapenemase *K. pneumoniae* isolates, and none of them were susceptible to ceftazidime–avibactam. Similar results were obtained in this study, including 15 multi-carbapenemase *K. pneumoniae* strains.

According to research conducted in US medical centers between 2018 and 2022 [34], 97.1% of 7153 *K. pneumoniae* isolates were susceptible to meropenem–vaborbactam. Another study from Pittsburgh [32] noted that the susceptibility of carbapenem-resistant *K. pneumoniae* strains to meropenem–vaborbactam reached 99.0%, with MICs ranging from ≤0.015 mg/L to >32 mg/L. The majority of these strains (39%) were KPC-positive. Gaibani et al. [35] noted that 87% of KPC-positive *K. pneumoniae* strains isolated from bloodstream infections in a hospital in Bologna were susceptible to meropenem–vaborbactam. In this study, over 30% of carbapenemase-positive *K. pneumoniae* strains were susceptible to meropenem–vaborbactam. Among the KPC-positive strains, all were susceptible to this drug. In contrast, Bianco et al. [33] analyzed susceptibility to selected antimicrobials in multi-carbapenemase *K. pneumoniae* isolates. Two out of seven strains were susceptible to meropenem–vaborbactam. However, in this study, which included 15 multi-carbapenemase isolates, none of the analyzed *K. pneumoniae* strains were susceptible to meropenem–vaborbactam.

None of the analyzed *K. variicola* strains produced carbapenemases, but there are articles in the literature about carbapenemase-positive *K. variicola* strains that produce NDM, KPC, IMP, or OXA enzymes [36,37,38,39]. Although there are a few reports in the literature assessing the susceptibility of *K. variicola* isolates to antimicrobials, they did not include fosfomycin or new combinations of β-lactams with β-lactamase inhibitors. The limited number of articles on the susceptibility of *K. variicola* to antimicrobials may be related to the challenges in identifying this species using automatic methods. Additionally, in the mass spectrometry method, the identification result depends on the number and type of spectra collected in the virtual library, making misidentification to other species within the complex possible.

In the era of increasing bacterial resistance to antimicrobials, rapid identification of carbapenemase-positive strains is a microbiological, clinical, and epidemiological concern. The emergence of MDR isolates often prompts difficult choices, such as using drugs with reduced susceptibility or increasing the drug dose. Therefore, it is crucial to monitor the emergence of resistant strains and quickly detect resistance to both “old” and “new” antimicrobials.

However, this study has some limitations. First, all strains were isolated from patients with symptoms of infection, and antimicrobial susceptibility testing was performed in vitro, making it unknown whether the same results would be obtained in vivo. Second, the strains came from patients in one hospital. Third, all *K. variicola* isolates in the study exhibited only an ESβL phenotype. Considering these aspects, further research should be continued and its scope expanded.

## 5. Conclusions

MDR *K. pneumoniae* complex infections, particularly those involving carbapenemase-producing strains, are becoming increasingly common. Fosfomycin, colistin, ceftazidime–avibactam, and meropenem–vaborbactam appear to be promising antimicrobials for treating infections caused by the MDR and PDR *K. pneumoniae* complex. The results obtained in this study underscore the importance of identifying carabapenemase. The most prevalent mechanisms of carbapenem resistance were NDM and OXA-181 enzymes.

Based on the obtained results, the in vitro activity of fosfomycin is quite satisfactory against VIM-positive *K. pneumoniae* isolates and ESβLs-positive *K. variicola* isolates. Colistin demonstrated the highest in vitro antimicrobial activity against strains producing KPC and VIM carbapenemases. However, new combinations of β-lactams with β-lactamase inhibitors (ceftazidime–avibactam and meropenem–vaborbactam) presented excellent in vitro activity against EsβLs- and KPC-positive isolates. Currently, strains producing more than one carbapenemase pose a significant challenge. The findings of this study indicate that only colistin showed good, but unsatisfactory, in vitro activity against *K. pneumoniae* complex strains. Therefore, the susceptibility of strains to any antimicrobial agents is not constant over time and should be monitored continuously.

## Figures and Tables

**Figure 1 antibiotics-13-00126-f001:**
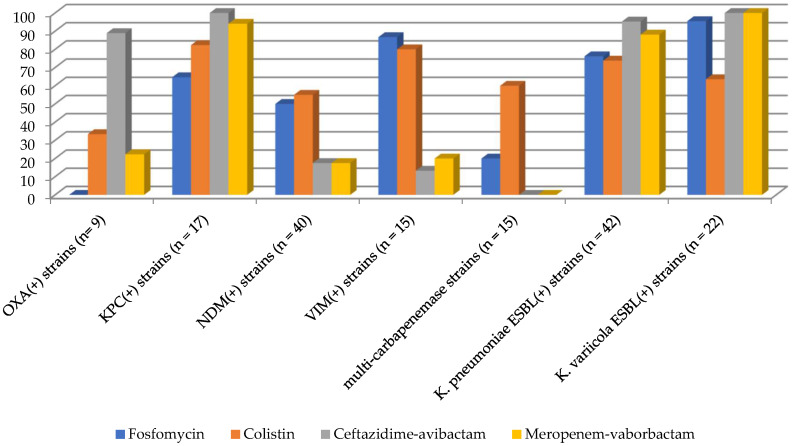
Susceptibility to selected antimicrobials of *K. pneumoniae* complex (*n* = 160).

**Table 1 antibiotics-13-00126-t001:** β-lactamases produced by *K. pneumoniae* complex strains (*n* = 160).

Enzymes Produced by *Klebsiella pneumoniae* Complex Strains (*n* = 160)
OXA-48-, CTX-M1-positive *K. pneumoniae* strains (*n* = 2)
OXA-181-, CTX-M1-positive *K. pneumoniae* strains (*n* = 7)
KPC-, CTX-M1-positive *K. pneumoniae* strains (*n* = 12)
KPC-positive *K. pneumoniae* strains (*n* = 5)
NDM-, CTX-M1-positive *K. pneumoniae* strains (*n* = 33)
NDM-positive *K. pneumoniae* strains (*n* = 7)
VIM-, CTX-M1-positive *K. pneumoniae* strains (*n* = 7)
VIM-positive *K. pneumoniae* strains (*n* = 8)
NDM-, OXA-181-, CTX-M1-positive *K. pneumoniae* strains (*n* = 13)
NDM-, OXA-181-, CTX-M1-, CTX-M9-positive *K. pneumoniae* strains (*n* = 1)
VIM-, NDM-, CTX-M1-positive *K. pneumoniae* strains (*n* = 1)
ESβLs-positive *K. pneumoniae* strains (*n* = 42)
ESβLs-positive *K. variicola* strains (*n* = 22)

**Table 2 antibiotics-13-00126-t002:** MIC_50_ and MIC_90_ (mg/L) values for selected antimicrobials.

Antimicrobial	*K. pneumoniae* Carbapenemase-Positive Strains (*n* = 96)		*K. pneumoniae* ESβLs-Positive Strains (*n* = 42)		*K. variicola* ESβLs-Positive Strains (*n* = 22)	
	MIC_50_	MIC_90_	MIC_50_	MIC_90_	MIC_50_	MIC_90_
Fosfomycin	24	48	12	32	4	16
Colistin	0.75	8	0.5	4	1.5	4
Ceftazidime- avibactam	16	128	0.75	2	0.5	0.5
Meropenem- vaborbactam	24	128	0.25	1.5	0.25	0.5

**Table 3 antibiotics-13-00126-t003:** MIC distribution for fosfomycin.

MIC Value (mg/L)	*K. pneumoniae* Carbapenemase-Positive Strains (*n* = 96)	*K. pneumoniae* ESβLs-Positive Strains (*n* = 42)	*K. variicola* ESβLs-Positive Strains (*n* = 22)
4	3	4	5
8	5	6	4
16	16	12	6
32	24	10	6
48	1	0	0
64	17	2	0
128	18	4	1
256	7	1	0
>256	5	3	0

MIC value according to EUCAST breakpoints 2023 (v. 13.0) [17].

**Table 4 antibiotics-13-00126-t004:** MIC distribution for colistin.

MIC Value (mg/L)	*K. pneumoniae* Carbapenemase-Positive Strains (*n* = 96)	*K. pneumoniae* ESβLs-Positive Strains (*n* = 42)	*K. variicola* ESβLs-Positive Strains (*n* = 22)
0.25	2	1	4
0.5	29	20	5
1	25	8	2
2	3	2	3
8	11	6	8
16	13	5	0
>16	12	0	0

MIC value according to EUCAST breakpoints 2023 (v. 13.0) [17].

**Table 5 antibiotics-13-00126-t005:** MICs distribution for ceftazidime–avibactam and meropenem–vaborbactam.

MIC Value (mg/L)	*K. pneumoniae* Carbapenemase-Positive Strains (*n* = 96)		*K. pneumoniae* ESβLs-Positive Strains (*n* = 42)		*K. variicola* ESβLs-Positive Strains (*n* = 22)	
	CZA	MV	CZA	MV	CZA	MV
0.032	0	1	0	1	0	1
0.047	0	1	1	2	2	0
0.064	0	1	0	1	1	2
0.094	0	1	0	1	0	3
0.125	0	0	0	6	0	0
0.19	0	0	0	2	0	0
0.25	0	1	5	0	0	4
0.38	1	0	1	3	4	2
0.5	2	1	3	6	5	4
0.75	3	1	3	1	4	3
1	4	4	12	3	3	1
1.5	11	6	7	0	0	1
2	8	6	6	6	3	1
3	1	1	0	2	0	0
4	4	2	2	3	0	0
8	1	3	1	2	0	0
12	0	0	0	1	0	0
16	2	2	0	0	0	0
32	4	2	0	0	0	0
48	1	8	0	0	0	0
64	3	5	1	0	0	0
96	1	2	0	0	0	0
128	6	3	0	0	0	0
256	16	18				
>256	28	27	0	2	0	0

MIC value according to EUCAST breakpoints 2023 (v. 13.0) [17], CZA: ceftazidime–avibactam, MV: meropenem–vaborbactam.

## Data Availability

The data presented in this study are available on a reasonable request from the corresponding author.

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
