# Peer review of "In Vitro Activity of “Old” and “New” Antimicrobials against the Klebsiella pneumoniae Complex"

_antibiotics, 2024, doi:10.3390/antibiotics13020126_

Round 1

Reviewer 1 Report

Comments and Suggestions for Authors

The authors evaluated the fosfomycin, colistin, ceftazidime-avibactam and meropenem-vaborbactam in vitro activity against multidrug-resistant K. pneumoniae complex strains by agar dilution method and microdilution method. Overall, the content of the article is too simplistic, the amount of data is too small, and there is not much valuable information. Moreover, the sample size is too small, and the authors didn’t mention whether the isolates were collected during a certain time period or for a long time, therefore, the representativeness of the strains is limited.

Comments on the Quality of English Language

The english is acceptible.

Reviewer 2 Report

Comments and Suggestions for Authors

This ms reports in vitro antibacterial activity of 4 antibiotics/combinations against Klebsiella pneumoniae complex from Poland. The findings provide confirmatory data for regional reference. The ms can be further improved for avoiding misleading clinicians by considering the limitations of this in vitro study. Comments are given below:

1. Title/L344. “last chance”: hard to comprehend. In the field, “last resort” is typically used. However, the antibiotics tested are not formally recognized as the last resort antibotics against K. pneumoniae. Clarity is needed from the author.

2. L78-79. The rationale to test 4 antibiotics/combinations are inadequate and can be improved for including additional descriptions. No mention for them as “last chance” or “last resort” antibiotics.

3. L115-125. This section (“2.4 MIC determination”) describes 3 times “MICs ranged from … to..” (L117, L119 and L122), which would be interpretated as “Results”. Please verify if these ranges are merely testing drug concentration ranges, rather than the MIC results.

4. L124. Please include the reference for the EUCAST document version used for MIC interpretation. Because interpretation criteria can change with time, strongly suggest including the EUCAST clinical breakpoints (S/R) in the ms.

5. L139-140. How multidrug-resistant and pandrug-resistant are defined? Please include the information in Section 2.4.

6. L345. Conclusion. Several descriptions (L350 for Fosfomycin; L352 for colistin; L354 for β-lactam combinations) for possible clinical outcome require to draw much attention to avoid misleading. This study only deals with in vitro activity, which is not enough to predict clinical effectives outcome 9thus the drug choices). The sponsor should consider the limitation of the study in the Discussion and Conclusions for in vitro activity study only.

7. Supplementary Table. Clarity is needed as the title shows “The susceptibility to…” with the right column showing “Susceptibility”, which does not define the strains as Susceptible or Resistant based on clinical breakpoints used.

8. Others (editorial; examples only):

L32. Key words: Group antibiotics and organisms closely, i.e., place “meropenem-vaborbactam” before “colistin” (no need to be alphabetical).

L43/L46/L103/L156/L287/L290, etc. Use Greek “β” for “beta”. Make the change throughout the ms.

L66. Add “(EARSS)” after “System”.

L118. No need to repat the full spelling of MIC (already in L116).

L153. Please write these enzymes correctly (e.g., close some spaces and use the right dash; “-7” is which enzyme?).

L234. Add “classes” before “A to D”.

L288. Add “U.S.” (or “US” as used in L315) before “Food”.

L26/L28/L232/251/258/311/L319/L320/L: 9 “in turn” are noted. Would suggest using  its synonyms to replace in some cases.

L346. Obviously, there is no deed to have “Probably this is the first” under a very specific situation. Let readers to decide the merit of a study.

References: Revise for consistency. Lower case article titles; write correct organisms; italicize organisms, etc. Ref 10, delete or revise “Eucast”

Comments on the Quality of English Language

English expression can be further improved.

Reviewer 3 Report

Comments and Suggestions for Authors

The author has evaluated various last line antibiotic including:  colistin, fosfomycin, ceftazidime-avibactam and meropenem-vaborbactam in vitro activity against multidrug-resistant K. pneumoniae complex strains. The finding from this study provides crucial information and serves as a guide for treatment decisions in the treatment of infections caused by K. pneumoniae and ESBL-positive K. variicola strains

This study is fascinating, relevant, and well-written, and only minor revision are needed

 Comments to Authors

1.      There is no mention of ethical considerations, such as approval from an ethics committee

or informed consent for sample collection.

2.      Line 368: Please include the specific grant number or project name (if applicable)

3.      Line :172 figure caption should be place in the bottom instead in top of the figure

Reviewer 4 Report

Comments and Suggestions for Authors

SÄ™kowska reports on in vitro antimicrobial susceptibility of selected essential antibiotics against ESBL- and carbapenemase-producing Klebsiella pneumoniae complex isolates recovered from a Polish healthcare facility. Owing to the overall global importance of antimicrobial resistance, and the eminent role occupied by ESBL and carbapenem resistance in the crisis, the study of SÄ™kowska has a lot of merits – clinically and epidemiologically. This is because there are limited treatment options for infection with isolates exhibiting such traits. Besides, the report contains succinct background information, relevant literature have also been cited, and the study is novel enough, contributing valuable data to help address the global health crisis of antimicrobial resistance. The author need to, however, improve the manuscript by addressing the following queries:

1.     Lines 150 to 151 of the Results section belongs in the Introduction, and also needs rephrasing for clarity.

2.     The author may want to make use of a professional English language editing service provider to go through the manuscript thoroughly and address issues with grammar, a few of which are pointed out below:

a.     On Line 15, “antimicrobials resistance” needs to be rewritten as “antimicrobial resistance”.

b.     On Line 19, the author may want to be consistent with the use of the dash punctuation (–).

c.     On Lines 25 to 29, each “has” needs to be rewritten as “had”.

d.     The sentence within Lines 37 to 39 needs rephrasing for clarity.

e.     On Line 47, the semicolon needs to be changed to a colon.

Throughout the manuscript, a generous use of commas would greatly improve clarity.

Comments on the Quality of English Language

Moderate English language edits are needed.

The authors may want to make use of a professional English language editing service.

Round 2

Reviewer 1 Report

Comments and Suggestions for Authors

The authors have made extensive revisions, and the article is acceptable in the current  form.

Author Response

Thank you for your detailed review of the manuscript